# Metabolic modeling of sex-specific liver tissue suggests mechanism of differences in toxicological responses

Connor J. Moore[1], Christopher P. Holstege[2], Jason A. Papin[1,3,4] *

1 Department of Biomedical Engineering, University of Virginia, Charlottesville, Virginia, United States of America, 2 Department of Emergency Medicine, Division of Medical Toxicology, University of Virginia, Charlottesville, Virginia, United States of America, 3 Department of Biochemistry & Molecular Genetics, University of Virginia, Charlottesville, Virginia, United States of America, 4 Department of Medicine, Division of Infectious Diseases and International Health, University of Virginia, Charlottesville, Virginia, United States of America

* papin@virginia.edu

**Data Availability Statement:** Code to reproduce this analysis is available at (https://github.com/ConnorMoore1/Sex-Spec-Hepato). Gene expression data available under GSE130991, GSE36059, and GSE5281.

## Abstract

Male subjects in animal and human studies are disproportionately used for toxicological testing. This discrepancy is evidenced in clinical medicine where females are more likely than males to experience liver-related adverse events in response to xenobiotics. While previous work has shown gene expression differences between the sexes, there is a lack of systems-level approaches to understand the direct clinical impact of these differences. Here, we integrate gene expression data with metabolic network models to characterize the impact of transcriptional changes of metabolic genes in the context of sex differences and drug treatment. We used Tasks Inferred from Differential Expression (TIDEs), a reaction-centric approach to analyzing differences in gene expression, to discover that several metabolic pathways exhibit sex differences including glycolysis, fatty acid metabolism, nucleotide metabolism, and xenobiotics metabolism. When TIDEs is used to compare expression differences in treated and untreated hepatocytes, we find several subsystems with differential expression overlap with the sex-altered pathways such as fatty acid metabolism, purine and pyrimidine metabolism, and xenobiotics metabolism. Finally, using sex-specific transcriptomic data, we create individual and averaged male and female liver models and find differences in the pentose phosphate pathway and other metabolic pathways. These results suggest potential sex differences in the contribution of the pentose phosphate pathway to oxidative stress, and we recommend further research into how these reactions respond to hepatotoxic pharmaceuticals.

## Author summary

There is a male-bias in clinical testing of drugs and a disproportionate number of hepatotoxic events occur in women. Previous work uses gene-by-gene differences in biological sex to explain this discrepancy, but there is little focus on the systematic interactions of

**Funding:** Support for this project was provided by the National Institutes of Health (R01-DK132369 to J.P. and T32-GM145443 to C.M.). The funders had no role in study design, data collection and analysis, decision to publish, or preparation of the manuscript.

**Competing interests:** The authors have declared that no competing interests exist.

these differences. To this end, we use a combination of gene expression data and metabolic modeling to compare metabolic activity between the male and female liver and treated and untreated hepatocytes. We find several subsystems with differential activity in each sex, and when comparing these subsystems with those pathways altered by hepatotoxic agents, we identify several pathways that overlap. To explore these differences on a reaction-by-reaction basis, we use the same sex-specific transcriptomic data to contextualize the previously published Human1 metabolic model. In these models we find a difference in flux through the pentose phosphate pathway, suggesting a potential difference in response to oxidative stress. These findings can help guide future drug design, toxicological testing, and sex-specific research to better account for the entire human population.

## Introduction

Male subjects in both animal and human studies are disproportionately used for testing in toxicology studies [1,2]. This discrepancy leads to incorrect assumptions on female drug response as evidenced in the clinic where female patients are more likely than males to experience liver-related adverse events in response to xenobiotics such as acetaminophen, diclofenac, and isoniazid [3–5]. While gene expression differences between males and females have been extensively studied [6–8], little is known about how these changes in expression contribute to functional changes that result in this divergent clinical response.

Sexually differential metabolism is a key to understanding these responses [9]. Transcriptional profiling can provide insight into which metabolic genes are expressed and how associated functions may differ between males and females. A previous study has shown that male and female serum metabolic profiles have distinct signatures [10], but connecting differences in gene expression and cellular machinery cannot be done on a gene-by-gene basis. To analyze trends across thousands of genes and reactions, a systems-level approach is required.

Genome-scale metabolic models (GEMs) have emerged as a tool for an integrated analysis of the genome, transcriptome, and metabolome. A GEM is a mathematical representation of all known reactions, metabolites, and gene-protein-reaction mappings (GPRs) that characterize intracellular metabolism of a given cell type. The GPRs associate each gene with one or more reactions, so any changes in genome content or in transcription can be evaluated mechanistically with the model. Previous work has investigated sex differences in metabolism [11] and characterized drug-toxicity responses [12] using GEMs, but there is a gap in the mapping of sex differences in metabolism to toxicological responses. Transcriptomic data can therefore be used to inform the GEM about which reactions are upregulated or downregulated for a given sex or drug. These differences in gene expression can then be evaluated in the context of GPRs to describe how the flux of each reaction changes in these conditions.

Here, we present an analysis of male and female liver metabolism using tissue-specific and drug-specific gene expression data in the context of a human GEM. Liver-, kidney-, and brain-sourced transcriptomics from the Gene Expression Omnibus (GEO) are used to establish general and liver-specific sex differences in metabolism where the liver acts as the sexually dimorphic tissue of interest, the kidney represents a similarly sexually dimorphic tissue, and the brain functions as a known sexually monomorphic tissue [6]. We also compare metabolism between hepatocytes with and without exposure to drug using expression data from ToxicoDB [13] with particular interest in those metabolic pathways known to differ between the male- and female-sourced liver tissue. We then use the male- and female-sourced liver gene expression data to create individual and averaged male- and female-specific GEMs and use flux

sampling to illustrate differences in core metabolism and metabolites involved in the pentose phosphate pathway. Together, these results suggest that sexually dimorphic adverse event frequency may be driven by differences in response to oxidative stress.

## Results

### Reporting frequencies of adverse events differ by sex

We used the Food and Drug Administration's Adverse Event Reporting System (AERS), a collection of side-effect reports voluntarily sourced from health care professionals and the general public, to quantify sex-specific adverse event frequency (Fig 1). Each report includes information on the event, the sex and age of the patient, other drugs being used, and the time of the incident. We collected reports from this database using AERSMine [14], an application designed to improve the accessibility of AERS. Reports related to liver dysfunction were counted for each quarter from 2004 to 2021 and divided by the total amount of reports for that quarter to account for increasing reporting trends (Fig 1A). We found that not only did more total reports exist for female patients, but that female patients are consistently reported to experience liver-related adverse events more often than their male counterparts (Fig 1B). This result can be explained in part by a previous report of overall increased prescription drug consumption in women [15], so we next investigated sexual-dimorphic response by drug. We compared the ratio of reports for each drug for each sex, only including drugs with greater than 100,000 reports to ensure that the differential effect was robust and not due to small sample size. Unexpectedly, more drugs were disproportionately affecting men, but those that were reported more often in women tended to have a higher fraction of female-specific reports (Fig 1C). It is important to note, however, that not all drugs are used equally by both sexes. For example, alendronic acid is primarily prescribed for osteoporosis and tenofovir disoproxil is used as an HIV treatment, diseases with higher female and male incidence, respectively [16,17]. Frequently-used pharmaceuticals exhibiting a spectrum of responses indicates that sex is a relevant variable in determining hepatotoxicity, so understanding the sex-based metabolic signature of toxicity is an important aspect of this problem.

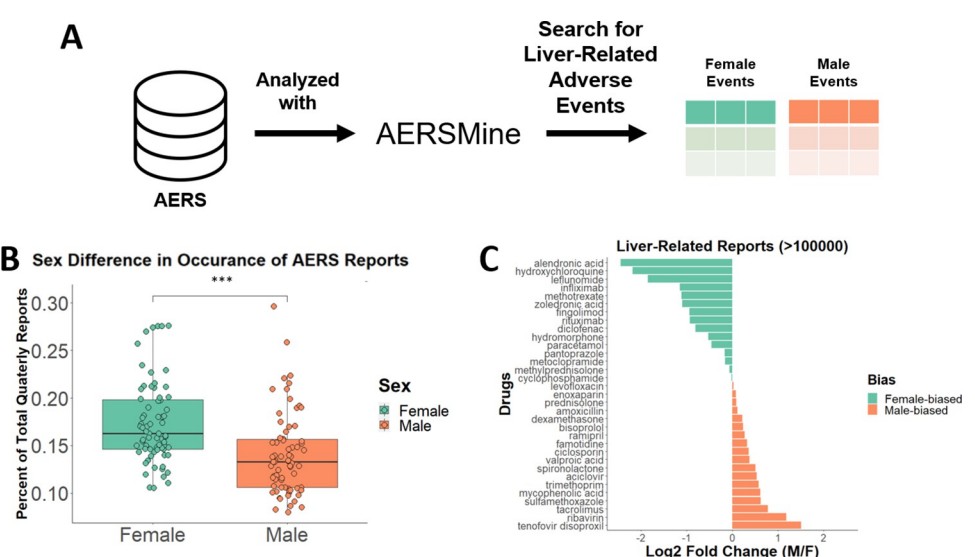

**Fig 1.** **(A)** The Adverse Event Reporting System (AERS) from the FDA was searched with AERSMine using liver-related terms. **(B)** All reports with sex data were counted for each quarter and compared. *** indicates significance of p-value < .001. **(C)** Reports were compared for the top reported drugs for all time periods.

### Transcriptome-informed metabolic model indicates a sex- and tissue-specific signature in untreated tissue

Next, we sought to understand differences in functional liver metabolic networks between males and females. To do this, we used microarray data from GEO to characterize gene expression differences between male and female patients not experiencing toxicity. To understand sexually dimorphic metabolism specific to the liver, kidney and brain tissue were also included in this analysis as a comparison. The kidney, as a tissue with sex-specific gene expression, allows us to identify those metabolic functions that may be sex-specific, but not tissue-specific, while the brain, as a notably sexually monomorphic organ, acts as a control [6].

For each tissue type, differentially expressed genes (DEGs) between the sexes were calculated (FDR < 0.1) (Fig 2A). We then used the publicly available GEM, Human1 [18], to provide context for the functional impact of the differentially expressed genes. Human1 accounts for the function of over 13000 reactions, 8300 metabolites, and 3600 genes with GPRs linking these genes and reactions. We applied Tasks Inferred from Differential Expression (TIDEs) [19] to the Human GEM to delineate metabolic systems that were differentially active. TIDEs has been used previously to identify metabolic tasks that are used differently; however, given the application here to various tissue type models without previously curated metabolic tasks, we apply TIDEs to subsystems of the network reconstruction (rather than metabolic tasks). The use of TIDEs to infer differentially active subsystems is different from other enrichment-type analyses given the application to a network reconstruction with GPR mappings that connect the genes to reactions. Using the subsystems defined by Human1, we can assign weights to each reaction in the subsystem dependent on the log fold change from the DEGs and the

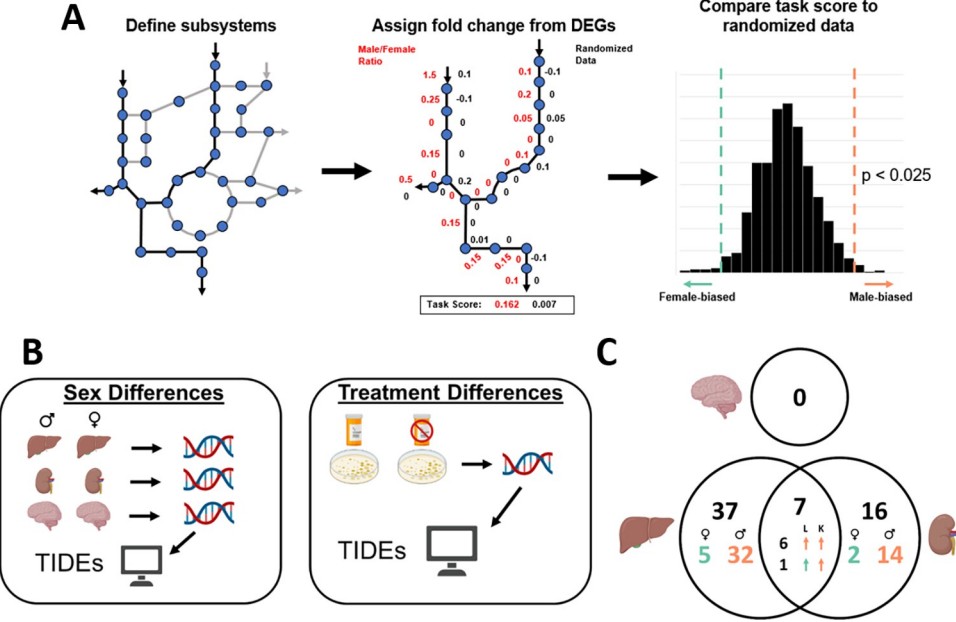

**Fig 2.** **(A)** Tasks Inferred from Differential Expression (TIDEs) is summarized. First, reactions are delineated as part of a metabolic subsystem. Expression data and GPRs are then used to assign log2 fold changes as weights to each reaction. The TIDEs score is then calculated as the average weight of the subsystem and compared to randomized fold changes from other genes in the data to determine significance. **(B)** This method is used to compare sex-specific untreated liver, kidney, and brain expression data as well as treated and untreated cultured hepatocyte expression data. **(C)** 37 subsystems are sex-biased exclusively in the liver, 16 are sex-biased exclusively in the kidney, and 0 are sex-biased in the brain. 7 subsystems are altered by sex in both the liver and kidney, with 6 upregulated in males in both tissue types, and 1 upregulated in the female liver and male kidney. Created with BioRender.com.

GPRs from the GEM to generate a TIDEs score for that subsystem. This TIDEs score can then be compared to randomized weights for each reaction to determine if the TIDEs score is significantly higher or lower (male- or female-biased) for that subsystem (Fig 2B, left). This score can therefore identify sex-biased metabolic subsystems by not only grouping together DEGs but by also considering how those DEGs, their subsequent proteins, and their corresponding reactions are related.

Of the 135 subsystems defined by the GEM, 37 exclusively in the liver tissue, 16 exclusively in the kidney, 7 in the liver and kidney, and 0 in the brain were found to be significantly different between the male and female patients (Fig 2C). The gene expression data for the male and female brain tissue did not have differences in TIDEs, suggesting highly similar metabolic function of the sexes in this tissue type. The expression profiles for the liver and kidney tissue, however, did have significant sex-biased TIDEs. Of the sex-biased subsystems, 7 were found to be biased in both kidney and liver tissues with 6 as male-biased (acylglycerides metabolism, cholesterol metabolism, glycine/serine/threonine metabolism, protein assembly, steroid metabolism, and tyrosine metabolism), 0 as female-biased, and 1 as female-biased in the liver and male-biased in the kidney (pentose and glucuronate interconversions). Subsystems that are sex-biased only in the liver include several core metabolic pathways, such as glycolysis and gluconeogenesis, fatty acid pathways, such as beta oxidation and activation, and nucleotide metabolism, including purine and pyrimidine metabolism. Each of these subsystems was found to be male-biased except purine and pyrimidine metabolism which both showed more activity in the female liver. To provide additional context to these pathways, we next look to how they are affected by the presence of hepatotoxicants.

## Hepatocyte expression data suggests some sex-biased subsystems are also altered by hepatotoxic pharmaceuticals

Using gene expression data from ToxicoDB [13], we used TIDEs to provide context to the DEGs between the treatment and no treatment conditions in male and female primary human hepatocytes. Looking only at those subsystems which were sex-biased in the liver, we found several subsystems that were also broadly affected by hepatotoxicants (Figs 3 and S1). Genes associated with fatty acid activation, fatty acid desaturation, vitamin B12 metabolism, and xenobiotics metabolism saw upregulation in the presence of these substances, while beta-oxidation of fatty acids were downregulated. We also found lesser but consistently affected subsystems in acylglycerides metabolism, cholesterol biosynthesis, purine metabolism, and pyrimidine metabolism. Additionally, fewer differences were found in the male-sourced hepatocytes compared to the female-sourced cells, but this result is likely due to the small number of drugs tested on the male-sourced hepatocytes. These subsystems that are altered by sex and treatment suggest that metabolism plays a role in the sex-specific response to hepatotoxic pharmaceuticals.

## Flux sampling of individual sex-specific models reveals intra- and inter-sex variability

While a subsystem-level view can help explain broad differences in metabolism, a more refined comparison of metabolism may provide additional insight. To understand how male and female liver metabolism differ on a reaction-by-reaction basis, we used the same liver-specific gene expression data from TIDEs with Reaction Inclusion by Parsimony and Transcript Distribution, or RIPTiDe [20]. RIPTiDe utilizes gene transcript abundances and overall flux parsimony to prune reactions that do not reflect the cell's transcriptome in a given context, in this case male or female liver (Fig 4A). With this method, we used each individual transcriptome to

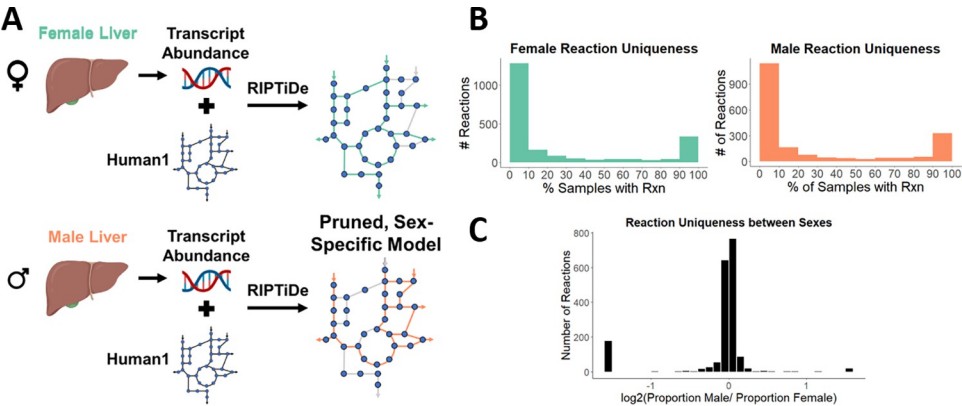

**Fig 3.** The results of a TIDEs score comparison of male- and female-sourced non-treated hepatocytes and drug-treated hepatocytes with TIDEs are shown with each row representing a different treatment and each column representing a different subsystem. Each subsystem was found to be sex-biased in our TIDEs analysis. We find several of these subsystems also have an altered behavior in response to known hepatotoxicants.

**Fig 4.** **(A)** RIPTiDe can be used to create sex-specific models from transcriptomic data and an existing model by mapping gene transcript abundance to reactions and pruning those that lack evidence of expression. **(B)** Reactions are checked for uniqueness within each sex. The number of sex-specific samples with a specific reaction is divided by the total number of samples for that sex, and that reaction is placed in the appropriate bin. This process is repeated for every reaction in each set of samples. **(C)** The proportion of male samples that contain a specific reaction was compared to the corresponding proportion contained in female samples to determine uniqueness between sexes. Bars at either extreme represent reactions specific to each sex. Created with BioRender.com.

**Table 1. Top Unique Subsystems based on Number of Unique Reactions in Female Models.**

| Subsystem | # of Unique Female Reactions |
|---|---|
| Nucleotide metabolism | 44 |
| Purine metabolism | 28 |
| Pyrimidine metabolism | 19 |
| TCA and glyoxylate/dicarboxylate metabolism | 16 |
| Glycolysis/Gluconeogenesis | 12 |

create sample-specific models. We tested various objective fractions and removed those models which could not achieve at least 40% of the original maximum biomass flux, leaving 403 female and 109 male models to analyze. Each model was sampled 110 times to reach the required sampling depth required for RIPTiDe.

An analysis of variability within each sex reveals reactions are frequently either a part of the core metabolism or are present in very few of the models (Fig 4B). A similar distribution can also be seen when comparing between sexes, where most reactions are found in near-equal proportions in each sex with the second largest group being sex-specific reactions (Fig 4C). Specifically, we found that the male and female models contained 18 and 177 unique reactions, respectively, that are in at least 10% of the samples for each sex. These unique reactions tended to group into specific pathways: nucleotide and core metabolism in females (Table 1) and transport reactions for males (Table 2), with only 3 subsystems unique to the male model. These analyses show that while there are differences in gene expression between individuals, there are several key metabolic pathways that differ between male and female livers as well.

## Differences in the pentose phosphate pathway suggest sex-specific differences in redox balance

We used RIPTiDe with averaged gene expression to create a male and female metabolic model. We found that the male model directly exports ribose while the female model first converts ribose to ribitol, producing NAD+ as well (Fig 5). This reaction is a key step in the pentose phosphate pathway (PPP) which is known to directly impact glycolysis, gluconeogenesis, nucleotide metabolism, and fatty acid metabolism which were all observed as sex-specific subsystems as described above. The PPP has well known redox homeostasis capabilities [21], and this result suggests sex differences in the pathway may result in differences in oxidative stress handling. Since oxidative stress is a notable outcome of drug toxicity [22], these sex differences found in the PPP may contribute to the sex-specific hepatotoxic response seen in the clinic.

## Discussion

We present an analysis of female-biased adverse event reporting, the sexually dimorphic profile of liver metabolism, the hepatocellular response to toxic pharmaceuticals, and differences in sex-specific GEMs. We find that sex as a biological variable is critical to describe the flow of metabolites through the hepatocyte and that there is some overlap between those subsystems

**Table 2. Top Unique Subsystems based on Number of Unique Reactions in Male Models.**

| Subsystem | # of Unique Male Reactions |
|---|---|
| Transport reactions | 16 |
| Carnitine shuttle (mitochondrial) | 1 |
| Fatty acid oxidation | 1 |

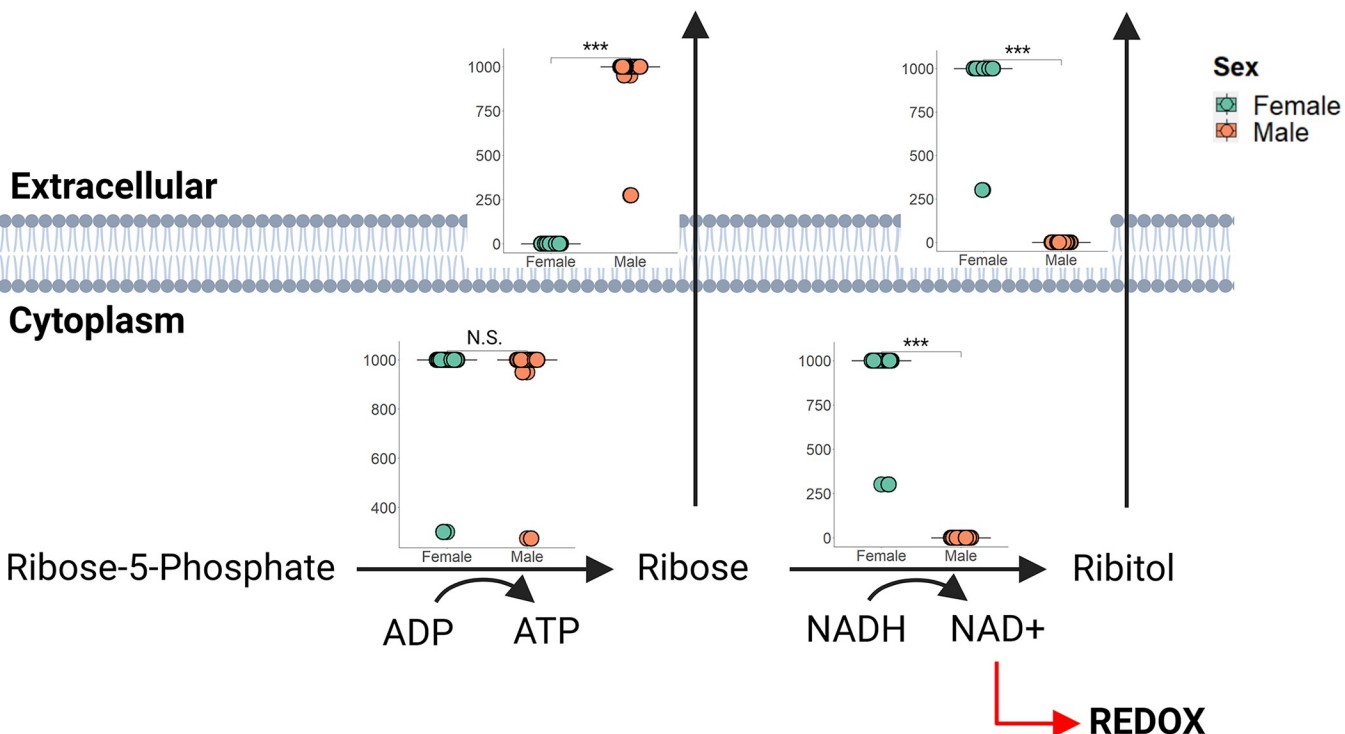

**Fig 5.** A selection of reactions from the pentose phosphate pathway is shown here. Ribose-5-phosphate is transformed into ribose equally by both models, but the female model first converts this ribose to ribitol before excretion. This conversion also produces a NAD+ molecule which can be used in redox metabolism. *** indicates significance of p-value < .001. N.S. indicates not significant. Created with BioRender.com.

which exhibit sex differences and those which experience changes in response to drug. Of the 135 subsystems defined in the Human1 metabolic network reconstruction, 44 in the liver, 23 in the kidney, and 0 in the brain were found to differ based on sex, suggesting that the liver may have more sex-specific differences in functional metabolism than other tissue types, supporting previous conclusions found in mice [6]. It is, however, important to note that in the human liver data set we used for our analysis, the participants are all classified as obese and therefore may be experiencing liver-related health issues that dilute the effect of sex, leading to potentially confounding results.

In the liver, we find several sex-biased subsystems of interest. Xenobiotic metabolism was more active in untreated males, while pentose and glucoronate interconversions were female-biased. This result points to a difference in pretreatment gene expression, which may result in different initial responses of phase I and phase II metabolism to hepatotoxic drugs. There is disagreement in previous reports about which enzymes and subsystems are upregulated in each sex [23–25], and an up-to-date, focused investigation of these properties could enable the prediction of sex-specific hepatotoxicity as well as suggest sex-specific therapeutic dosing in clinical practice. We also observe sex-bias in bile acid biosynthesis. In combination with xenobiotic metabolism, this result may suggest differences in bacterial deconjugation [26] driven by sex differences in the gut microbiome [27]. We also find differences in several essential metabolic pathways, such as glycolysis/gluconeogenesis, nucleotide metabolism, and lipid metabolism with supporting evidence in human [28] or rat [29,30] hepatocytes.

Our analysis into the response of hepatocytes to cytotoxic drugs also showed changes in lipid metabolism, nucleotide metabolism, and xenobiotic metabolism. Though dysregulated

lipid metabolism [31], decreased DNA replication [32], and upregulated xenobiotics metabolism are notable markers of hepatotoxicity, we also found vitamin B12 to be altered by treatment, which we only identified in the literature for one drug [33]. We saw a small difference in activity in those cells sourced from male and female tissue, but this result is likely explained by the small number of drugs tested on the male hepatocytes. Additionally, this expression data was collected from primary hepatocytes which have been shown to lose hepatic functionality as time from collection increases, though this method is still considered the "gold standard" for *in vitro* human hepatocyte testing [34]. Despite this caveat, we find it interesting that several of these subsystems were altered by sex and drug treatment and suggest further investigation into these groups of reactions.

Our individual models suggest the presence of a core and unique metabolism both within each sex and between the sexes. Between the sexes, this unique metabolism involved mainly transport reactions in males and nucleotide metabolism and core metabolism in females. Purine metabolism, pyrimidine metabolism, nucleotide metabolism, and glycolysis/gluconeogenesis were all found to be sex-biased TIDEs as well, with purine and pyrimidine metabolism also showing altered activity in response to hepatotoxicants. Because previous literature suggests nucleotide synthesis, lipid synthesis, and glycolysis can be impacted by the PPP [21], we chose to investigate differences in this pathway through our averaged sex-specific liver models. We found differences in ribose and ribitol export with the female model transforming ribose to ribitol before export while producing NAD+ in the process. The PPP has also been shown to be a "metabolic redox sensor" [35], a key to neuroinflammation in humans [36], and a mediator of oxidative stress [37]. Oxidative stress and inflammation of notable markers of drug induced injury [32], suggesting that the PPP may exhibit sex-specific activity in response to drugs and may be contributing to the sexually dimorphic response we see in the clinic.

There are important factors to note when considering the results of this analysis. First, there are several known limitations when using the AERS database including unknown prescription trends, voluntary reporting, and physician-bias in reporting certain drugs [38]. Though the analysis we provide here is limited, we hope it emphasizes the importance of sex in hepatotoxicity. Also, an assumption made with many gene expression data integration algorithms is that expression levels are related to protein levels and subsequently the presence or absence of a specific metabolic reaction. Certainly, there are post-transcriptional and post-translational modifications that can alter protein abundance and affect metabolic activity [39]. Though this assumption is not unique to this paper, it is nonetheless an important caveat to many gene expression integration methods. Additionally, our analysis here only considers one tissue type at a time; the cross-talk between organs and organ systems is a necessary consideration when evaluating toxicity as absorption, distribution, metabolism, and excretion by different organs can impact liver injury [40].

Sex as a biological variable continues to be a relevant consideration for any biological study. Many expression datasets do not have sex information recorded with the samples, decreasing the quality of the data as well as our pool of potential data that can be modeled. An understanding of the necessity for sex as a variable will provide the foundation required to further the pursuit of precision medicine and drug therapy.

## Methods and materials

### Adverse event reporting

We used AERSMine [14] to evaluate the difference in adverse event reporting in the United States between the first quarter of 2004 to the third quarter of 2021. Only those reports with age and sex information were used, and patients between the ages of 15 to 65 were considered. Adverse events were considered "liver-related" if labelled with one of the following: "drug-

induced liver injury", "hepatotoxicity", "hepatic enzyme increased", "hepatic and hepatobiliary disorders nec", "liver function analyses", "hepatic and hepatobiliary disorders", "hepatic failure and associated disorders", and "hepatic enzymes and function abnormalities". Once compiled, the percentage of total reports was calculated for each quarter and compared with a two-sided Mann-Whitney U test. Visualization was performed in R.

### Gene expression analysis

Differential gene expression between male and female samples from data sets were found in the Gene Expression Omnibus (Liver: GSE130991, Kidney: GSE36059, Brain: GSE5281). GSE130991 contains 910 liver samples from obese patients with and without statins; GSE36059 contains 403 kidney samples from tissue that was to be transplanted; and GSE5281 contains 161 brain samples from Alzheimer's Disease and control patients. The liver data set (GSE130991) was further filtered for those patients not on statins between the ages of 18 and 50. For those datasets without information on sex, male was assigned to those samples that were in the top 16% of the male-specific SRY gene and female to those in the bottom 16% (one standard deviation above and below the mean). Differential gene expression data for treated primary hepatocytes was downloaded from the ToxicoDB website (https://www.toxicodb.ca/) for all compounds categorized as "Most-DILI" for the 24-hour timepoint. Differentially expressed genes were then found using the limma package [41] and filtered to those genes that could be identified in the Human1 metabolic network reconstruction. Those genes that were significantly different between male and female were assigned their log2 fold difference, and all other genes were assigned a log2 difference of 0.

### Biased metabolic subsystems

To identify the presence of sex-biased metabolic subsystems, Tasks Inferred from Differential Expression (TIDEs) [19] and Human1 [18] were used. TIDEs identifies all genes related to reactions in a user-defined "task" (or subsystem) and assigns each reaction a weight based on the log2 fold difference in the expression of those genes. For genes with an "OR" gene protein rule, the highest fold difference will be used; for genes with an "AND" rule, the lowest will be used. Each fold difference is then averaged for a given subsystem, and this average becomes its TIDEs score. This score is then compared to 1000 randomized task scores, calculated using randomly chosen log fold difference weights from other subsystems. Significance was decided if $p < 0.025$ because it is a two-sided test. Subsystems were defined as KEGG ortholog subsystems as assigned by the model Human1.

### Sex-specific models

Sex-specific liver models were created using Human1 [18], gene expression data (GSE130991), and Reaction Inclusion by Parsimony and Transcript Distribution (RIPTiDe) [20]. Using gene expression data, RIPTiDe assigns linearly distributed weights between 0 and 1 to each reaction in Human1 with 0 being the reaction assigned the highest abundance gene transcript. The sum of fluxes is then minimized as the objective function, and any reactions with 0 flux are pruned. The weights in the resulting model are reassigned inverse to the previous (1 referring to the reaction with the highest gene abundance) and the sum of fluxes is maximized. The model is sampled using these constrained flux distributions. This technique was performed with male and female liver data, with 512 (403 female/109 male) individual models and averaged models for male and female were created and sampled.

## Supporting information

**S1 Fig. Heatmap with drug-induced TIDEs scores including specific drugs used for each row.** The results of a TIDEs score comparison of male- and female-sourced non-treated hepatocytes and drug-treated hepatocytes with TIDEs are shown with each row representing a different treatment and each column representing a different subsystem. Each rectangle represents the TIDEs score for a subsystem (column) when treated with a drug (row). Each subsystem was found to be sex-biased in our TIDEs analysis. We find several of these subsystems also have an altered behavior in response to known hepatotoxicants.
(DOCX)

## Author Contributions

**Conceptualization:** Connor J. Moore, Christopher P. Holstege, Jason A. Papin.

**Data curation:** Connor J. Moore.

**Formal analysis:** Connor J. Moore, Jason A. Papin.

**Funding acquisition:** Connor J. Moore, Jason A. Papin.

**Investigation:** Jason A. Papin.

**Methodology:** Connor J. Moore, Jason A. Papin.

**Project administration:** Jason A. Papin.

**Resources:** Jason A. Papin.

**Supervision:** Jason A. Papin.

**Validation:** Connor J. Moore.

**Visualization:** Connor J. Moore.

**Writing – original draft:** Connor J. Moore, Jason A. Papin.

**Writing – review & editing:** Connor J. Moore, Christopher P. Holstege, Jason A. Papin.

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
