## [Decision Letter · Decision Letter 0]

6 Mar 2023

Dear Professor Papin,

Thank you very much for submitting your manuscript "Metabolic modeling of sex-specific tissue predicts mechanisms of differences in toxicological responses" for consideration at PLOS Computational Biology.

As with all papers reviewed by the journal, your manuscript was reviewed by members of the editorial board and by several independent reviewers. In light of the reviews (below this email), we would like to invite the resubmission of a significantly-revised version that takes into account the reviewers' comments.

We cannot make any decision about publication until we have seen the revised manuscript and your response to the reviewers' comments. Your revised manuscript is also likely to be sent to reviewers for further evaluation.

Sincerely,

Kiran Raosaheb Patil, Ph.D.

Section Editor

PLOS Computational Biology

Kiran Patil

Section Editor

PLOS Computational Biology

Reviewer's Responses to Questions

**Comments to the Authors:**

Reviewer #1: In this study, the authors utilize GEM to analyze sex-dependent differences between female and male with regard to metabolic alteration in toxicological responses mostly in liver, but also in kidney and brain. The authors identified several metabolic pathways (subsystems) that are dysregulated in liver between male and female. In general, the manuscript is well written; methods and results are clearly presented and sufficiently discussed. This study provides insight into how sex affects drug response in the treatment of liver diseases, which is an important step toward precision medicine. I have number of minor comments.

1. Line 149 – 150. It is unclear if the author is saying that the two metabolic pathways were upregulated in males in both liver and kidney, or either in liver or in kidney (when using the term “between the liver and kidney”).

2. Line 145. The authors reported that 11 metabolic subsystems were dysregulated in liver. However, in the following text the authors only mentioned two (acylglyceride and steroid) and the other eight metabolic subsystems in Figure 2C.

3. Line 154 – 171. Most of the content seems like discussions rather than results. I suggest the structuring of the results and discussions carefully.

4. Line 175 – 176. As this is the first sentence, it is better to indicate what cells/materials were used in addition to the words “treatment and non-treatment conditions”.

5. Line 155. I guess here it should be Figure 2C not Figure 3A.

Reviewer #2: Please see attachment

Reviewer #3: Summary

The authors Integrate gene expression data to contextualize the previously published Human1 human cell metabolic network model to characterize the impact of transcriptional changes of metabolic genes in the context of sex differences and drug treatment.

They used Tasks Inferred from Differential Expression (TIDEs) to discover that androgen, ether lipid, glucocorticoid, tryptophan, and xenobiotic metabolism have more activity in the male liver, and serotonin, melatonin, pentose, glucuronate, and vitamin A metabolism have more activity in the female liver. On drug treatment in hepatocytes the largest differences berween the sexes are in subsystems related to lipid metabolism. sex-specific transcriptomic data was used to create individual and averaged male and female liver models. The results indicate that the sexually dimorphic behavior of the liver may be caused by differences in enterohepatic recirculation. The authors suggest an investigation into sex-specific microbiome composition for further understanding. Moreover, the authors point out the male-bias in clinical testing of drugs that has unfortunately led to a disproportionate number of hepatotoxic events in women and reiterate the lack of focus on the systematic interactions of these differences.

Comments:

There are no major issues with this study as detailed below. Based on the technical and scientific soundness of the study, I recommend publication.

(1) The question the authors are trying to answer is very relevant and the need of the hour

(2) Overall, the findings reported here can help shape sex-specific research to represent the entire human population and potentially guide future drug design and toxicological testing.

(3) There are some minor typographical errors and I recommend correcting them before publication

**Have the authors made all data and (if applicable) computational code underlying the findings in their manuscript fully available?**

Reviewer #1: Yes

Reviewer #2: **No: **A GitHub repository is available, but does not allow the reproduction of the analyses because the datasets are not available. Please see the review file for more detail.

Reviewer #3: Yes

PLOS authors have the option to publish the peer review history of their article (what does this mean?). If published, this will include your full peer review and any attached files.

Reviewer #1: No

Reviewer #2: No

Reviewer #3: No
---

## [Decision Letter · Decision Letter 1]

25 Jul 2023

Dear Professor Papin,

We are pleased to inform you that your manuscript 'Metabolic modeling of sex-specific liver tissue suggests mechanism of differences in toxicological responses' has been provisionally accepted for publication in PLOS Computational Biology.

Best regards,

Mark Alber, Ph.D.

Section Editor

PLOS Computational Biology

Kiran Patil

Section Editor

PLOS Computational Biology

Reviewer's Responses to Questions

**Comments to the Authors:**

Reviewer #1: The authors revised the paper based on the reviewer comments and I am satisfied with the latest version of the paper.

**Have the authors made all data and (if applicable) computational code underlying the findings in their manuscript fully available?**

Reviewer #1: Yes

PLOS authors have the option to publish the peer review history of their article (what does this mean?). If published, this will include your full peer review and any attached files.

Reviewer #1: **Yes: **Adil Mardinoglu

---

## [Editor Report · Acceptance letter]

17 Aug 2023

PCOMPBIOL-D-23-00166R1 

Metabolic modeling of sex-specific liver tissue suggests mechanism of differences in toxicological responses

Dear Dr Papin,

I am pleased to inform you that your manuscript has been formally accepted for publication in PLOS Computational Biology. Your manuscript is now with our production department and you will be notified of the publication date in due course.

With kind regards,

Zsofi Zombor
